# The influence of demographic and socio-economic factors on non-vaccination, under-vaccination and missed opportunities for vaccination amongst children 0–23 months in Kenya for the period 2003–2014

**Christopher Ochieng' Odero**[1]*, **Doreen Othero**[1], **Vincent Omondi Were**[2], **Collins Ouma**[3]

**1** Department of Public Health, Maseno University, Kisumu, Kenya, **2** KEMRI Wellcome-Trust Research Program, Health Economics Research Unit, Nairobi, Kenya, **3** Department of Biomedical Sciences and Technology, Maseno University, Kisumu, Kenya

* christodero@gmail.com

**Data Availability Statement:** This was a quantitative study that analyzed secondary data

## Abstract

Vaccination is crucial in reducing child mortality and the prevalence of Vaccine-Preventable-Diseases (VPD), especially in low-and-middle-income countries like Kenya. However, non-vaccination, under-vaccination, and missed opportunities for vaccination (MOV) pose significant challenges to these efforts. This study aimed to analyze the impact of demographic and socio-economic factors on non-vaccination, under-vaccination, and MOV among children aged 0–23 months in Kenya from 2003 to 2014. A secondary data analysis of data from the Kenya Demographic Health Surveys (KDHS) conducted during this period was conducted, with a total of 11,997 participants, using a two-stage, multi-stage, and stratified sampling technique. The study examined factors such as child's sex, residence, mother's age, marital status, religion, birth order, maternal education, wealth quintile, province, child's birth order, parity, number of children in the household, place of delivery, and mother's occupation. Binary logistic regression was employed to identify the determinants of non-vaccination, under-vaccination, and MOV, and multivariable logistic regression analysis to report odds ratios (OR) and their corresponding 95% confidence intervals (CI). In 2003, the likelihood of non-vaccination decreased with higher maternal education levels: mothers who did not complete primary education (AOR = 0.55, 95% CI = 0.37–0.81), completed primary education (AOR = 0.34, 95% CI = 0.21–0.56), and had secondary education or higher (AOR = 0.26, 95% CI = 0.14–0.50) exhibited decreasing probabilities. In 2008/09, divorced/separated/widowed mothers (AOR = 0.22, 95% CI = 0.07–0.65) and those with no religion (AOR = 0.37, 95% CI = 0.17–0.81) showed lower odds of non-vaccination, while lower wealth quintiles were associated with higher odds. In 2014, non-vaccination was higher among younger mothers aged 15–19 years (AOR = 12.53, 95% CI = 1.59–98.73), in North Eastern Province (AOR = 7.15, 95% CI = 2.02–25.30), in families with more than 5 children (AOR = 4.19, 95% CI = 1.09–16.18), and in children born at home (AOR = 4.47, 95% CI = 1.32–15.17). Similar patterns were observed for under-vaccination and MOV. This information can inform

from longitudinal repeated annual and national cross-sectional surveys obtained from the Kenya demographic and health surveys (KDHS) conducted in 2003, 2008/9, and 2014. The datasets are publicly available at https://dhsprogram.com. The relevant datasets regarding the children's information (Kids recode [KR]) were used. The STATA format was extracted and analyzed.

**Funding:** The authors received no specific funding for this work.

**Competing interests:** The authors have declared that no competing interests exist.

strategies for bridging the gaps in immunization coverage and promoting equitable vaccination practices in Kenya.

## Introduction

Immunization is the most cost-effective way to reduce child mortality in low and middle-income countries as it prevents 2–3 million deaths every year [1]. Vaccine-Preventable Diseases (VPD) such as measles, poliomyelitis, tuberculosis, tetanus and diphtheria are among the main causes of mortality and morbidity in children in developing countries. Vaccination coverage gaps like under-vaccination, non-vaccination and (MOV) are among the main threats to global health [2].

In Kenya, the National Vaccines and Immunization Program (NVIP) recommends that a child receives Bacillus Calmette–Guérin (BCG) and oral polio vaccine (OPV) at birth; pentavalent vaccine, OPV, rotavirus vaccine and pneumococcal conjugate vaccine (PCV) at 6, 10 and 14 weeks, vitamin A at 6 months and measles vaccine at 9 months [3]. Routine vaccines are delivered for free at public health facilities, with outreach services conducted in hard-to-reach communities. Through routine vaccination the government aims to achieve equitable access to health services, especially for children [3].

Non-vaccination and under-vaccination predispose children to VPDs which in most cases raise the cost of healthcare indirectly [4]. Non-vaccinated children tend to be characteristically uniquely different from under-vaccinated children and they tended to be clustered geographically, increasing their risk of transmitting VPDs to other non-vaccinated and under-vaccinated children [5]. Globally, close to 19.4 million children fail to receive the basic vaccination during their first year of life as a result of MOV [6]. In low-and-middle income countries MOV accounts for approximately one-third of children who visit health facilities [7], while specifically for Africa, it accounts for 1 out of every 4 children [8].

A child is said to be fully vaccinated if they have received all the basic vaccinations in the country's immunization schedule before their first birthday and under-vaccinated if they did not receive any of the recommended vaccines [9]. Kenya has also adopted the WHO protocols for assessing missed opportunity for vaccination (MOV) [7]. The MOV is defined as any contact with health facility that did not result in an eligible child receiving eligible vaccination [7]. The MOV may occur during curative or preventive services (e.g. oral rehydration training sessions growth monitoring and nutrition assessments). Reducing MOV is therefore critical in order to attain and sustain the 90% or more immunization coverage goal [10]. The common vaccines being missed are those given at birth and at six weeks of age (BCG,OPV0,OPV1, HBV1 and DTP1) [11].

Understanding of multi-level determinants that influence non-vaccination, under-vaccination and MOV is important in teasing out both individual and community level characteristics [12, 13]. Understanding the influence of demographic and socio-economic factors on non-vaccination, under-vaccination and MOV is necessary for implementing targeted strategies to ensure childhood immunization coverage gaps have been addressed and adequate policies implemented [14]. There are several studies that have been conducted within the East African region to understand the magnitude of non-vaccination, under-vaccination and MOV [15–20]. However, most of these studies did not delve into the influence of demographic or socio-economic factors on these immunization coverage gaps and how they change over time. Further, no studies have been conducted in Kenya using National level data to understand the

influence of demographic and socio-economic factors on these immunization coverage gaps. We therefore conducted this study whose objective is to understand the influence of demographic and socio-economic factors on non-vaccination, under-vaccination and MOV amongst children 0–23 months in Kenya for the period 2003–2014.

## Materials and methods

### Study design

This was a quantitative study that analyzed secondary data from longitudinal repeated annual and national cross-sectional surveys obtained from the Kenya demographic and health surveys (KDHS) conducted in 2003, 2008/9, and 2014. The datasets are publicly available at https://dhsprogram.com. To capture fundamental child health changes and policies related to vaccinations that have occurred over time, a 10-year trend analysis offered better insight into the possible trends in vaccination gaps.

### Study area

This study covered the entire 47 counties in Kenya. The KDHS surveys were national level surveys, which in 2003 and 2008/09 were conducted across all the former eight Provinces of Kenya and in 2014, was conducted in all the 47 counties in line with the new administrative units. The changes in the administrative units (Counties) were implemented to enhance administration and service delivery, including for health among others. The administrative units (Counties) were curved out of the former 8 provinces in Kenya. In our analysis, we have grouped the Counties as per the 8 former provinces to ensure consistency in reporting as per the KDHS data and evaluating trends across the years.

### Study population

The study population were children aged 0–23 months for each year of the KDHS studies.

### Sampling technique

The KDHS employed a comprehensive sampling methodology that encompassed a two-stage, multi-stage, and stratified approach, with households serving as the primary sampling unit. Within each selected household, women between the ages of 15 and 49 were interviewed. To gather a holistic set of data, the KDHS questionnaires incorporated a mother questionnaire, which gathered information on both mothers and their children.

To ensure that the findings accurately represented the entire population, sampling weights were meticulously designed. These weights took into account the uneven distribution of sampling probabilities and non-response rates. They played a crucial role in enabling weighted analyses that could be extrapolated to the entire population. The formulation of sampling weights aimed to address variations in the likelihood of selecting sampling units, households, and individuals, while also considering the response rate within different strata, which encompassed various geographical regions like counties and enumeration areas. Specifically, the household weight for a particular household was calculated by taking the reciprocal of its household selection probability and multiplying it by the reciprocal of the household response rate within the stratum. Subsequently, the individual weight was determined by multiplying the household weight by the reciprocal of the individual response rate within the same stratum.

An essential aspect of this approach was the categorization of households and individuals into sample strata to calculate response rates. This meticulous process guaranteed that the

analysis included a representative sample of the population and upheld the statistical integrity of the findings. For this analysis, complete child datasets from the years 2003, 2008/9, and 2014 were utilized.

## Sample size

The study used the KDHS sample size estimation process, which aimed to estimate the minimum number of women aged 15–49 years, number of households, number of children under five years and 12–23 months. In addition, in 2014, the sample size calculation was made to account for county level estimates in line with the devolved units. The sample size was estimated for each indicator, with varying standard error estimates, level of coverage and estimated response rates. Whilst the sample size for full immunization coverage of children aged 12–23 months was used in the determination of non-vaccination and under-vaccination, this study was restricted to children aged 0–23 months for all the years. The sample size for full immunization coverage of children aged 0–23 months in 2014 was as follows;

$$ n = Deft^2 \frac{(\frac{1}{p} - 1)}{\alpha^2} / (R_{i\,X}\, R_{h\,X}\, d) $$

Where;
 n–The sample size in households;
 Deft–The design effect of 1.800
 P–The estimated proportion (0.792)
 $\alpha$–the desired relative standard error; (SE = 0.011)
 $R_i$–The individual response rate; 92.6% (0.926)
 $R_h$–The household gross response rate; 98% (0.98)
 d–The number of eligible individuals per household.1.05
 Based on these assumptions, the sample size was 2380 in 2003, 2237 in 2008/2009 and 7380 in 2014.

## Data management and statistical analysis

The DHS data is publicly available and was obtained from the DHS program webpage for the period 2003, 2008/9, and 2014. The relevant datasets regarding the children's information (Kids recode [KR]) were used. The STATA format was extracted and analyzed. All analysis datasets and all reports generated from this data were stored in an access-controlled google drive, only accessible to the investigator. All coding and recoding's were done using Stata version 14.

The study had three outcome variables which were under vaccination, non-vaccination, and MOV. Under vaccination was defined as children who had missed out on at least one vaccine, non-vaccination referred to children who did not get any vaccine in the immunization schedule and MOV was defined as any contact with a health facility that did not result in an eligible child receiving vaccination.

Explanatory variables were selected based on prior knowledge and availability of variables in the datasets. They included; sex of child, residence, mother's age, marital status, religion, birth order, maternal education, wealth quintile, province, child's birth order, parity, number of children in the household, place of delivery and mother's occupation. The categorization was done according to the existing literature in the DHS reports.

To determine the level of non-vaccination, under-vaccination and missed opportunities for vaccination among children aged 0–23 months, a weighted descriptive analysis was used to estimate the prevalence of non-vaccination and under-vaccination, with 95% confidence

intervals (CI) for each year. The outcome variables were defined using an algorithm, as binary variables, and were coded as 1 if the child was either under-vaccinated, not vaccinated or had a MOV and 0 if they were vaccinated. The binary logistic regression technique was used to find the determinants of non-vaccination, under vaccination and MOV. For all comparisons, results with p-values <0.2 were included in a final multivariable logistic regression. This allowed for a more comprehensive exploration of potential factors influencing the outcome variables, besides those with statistically significant p-values ($\leq$0.05), ensuring that all relevant variables are considered while still maintaining the rigor of statistical significance in the final model. [21]. The odds ratio (OR) and 95% confidence interval were reported.

## Ethics approval and consent

Ethical review and approval were sought from the Maseno University Institutional Review Board/ Ethics Review Committee (IRB/ERC) before the start of study procedures. The team further sought research permit from the National Commission for Science, Technology and Innovations (NACOSTI). No informed consent was obtained before the secondary analysis. Consent provided during the KDHS typically includes a statement that allows for the data collected to be used for secondary analysis. To protect the confidentiality of the participants and their data, this secondary analysis used anonymized data.

## Results

### Socio-demographic characteristics of children aged 0–23 months in Kenya; 2003, 2008/09 and 2014

Table 1 provides a detailed overview of the socio-demographic characteristics of children aged 0–23 months in Kenya for the years 2003, 2008/09, and 2014. The data shows a nearly equal distribution of male and female children across all three years, with males slightly outnumbering females in 2003 (49.9% male, 50.1% female) and 2014 (50.9% male, 49.1% female). There was an increase in urban residence from 2003 to 2014, with a corresponding decrease in rural residence. In 2003, 18.8% of children lived in urban areas, which rose to 35.2% in 2014, while rural residence decreased from 81.2% to 64.8% over the same period. The majority of mothers were within the 20–29 age range, with slight variations observed over the years. In 2014, 29.3% of mothers were aged 20–24, and 29.1% were aged 25–29. The data further indicates that the majority of mothers were married or living together across all three years, with proportions remaining relatively stable. In 2014, 83.3% of mothers were married or living together. The largest religious group among respondents was Protestant/other Christian, though there was a slight decrease in this group by 2014. In 2014, 70.4% of respondents identified as Protestant/ other Christian. There was an improvement in maternal education levels over the years, with an increasing proportion of mothers attaining secondary education or higher. In 2014, 33.7% of mothers had completed secondary education or higher, compared to 20.8% in 2003. Wealth distribution remained relatively stable over the years, with minor fluctuations in proportions across quintiles. In 2014, 25.2% of respondents were in the lowest wealth quintile, compared to 24.8% in 2003. The Rift Valley consistently had the highest proportion of respondents, followed by Nyanza and Coast, with slight variations in proportions over the years. In 2014, Rift Valley accounted for 29.5% of respondents. The proportion of first-born children increased from 24% in 2003 to 26.6% in 2014, while the proportions of children with birth orders 2–4 and 5+ showed slight declines. Parity remained relatively stable over the years, with the majority of respondents falling within the 2–4 range. In 2014, 53.2% of respondents had a parity of 2–4. There was a decrease in the proportion of households with 0–1 children and an increase

**Table 1. Socio demographic characteristics of children aged 0–23 months in Kenya; KDHS 2003, 2008/09 and 2014.**

| Variable | 2003 | | 2008 | | 2014 | |
|---|---|---|---|---|---|---|
| **Sex of child** | n (2380) | % | n (2237) | % | n (7380) | % |
| Male | 1187 | 49.9 | 1157 | 51.7 | 3757 | 50.9 |
| Female | 1193 | 50.1 | 1081 | 48.3 | 3623 | 49.1 |
| **Residence** | | | | | | |
| Urban | 447 | 18.8 | 449 | 20.1 | 2597 | 35.2 |
| Rural | 1934 | 81.2 | 1788 | 79.9 | 4783 | 64.8 |
| **Mother's Age** | | | | | | |
| 15–19 | 283 | 11.9 | 211 | 9.4 | 706 | 9.6 |
| 20–24 | 742 | 31.1 | 719 | 32.1 | 2159 | 29.3 |
| 25–29 | 597 | 25.1 | 590 | 26.4 | 2150 | 29.1 |
| 30–34 | 413 | 17.4 | 423 | 18.9 | 1297 | 17.6 |
| 35–39 | 235 | 9.9 | 195 | 8.7 | 764 | 10.4 |
| 40–44 | 94 | 3.9 | 83 | 3.7 | 263 | 3.6 |
| 45–49 | 16 | 0.7 | 16 | 0.7 | 40 | 0.5 |
| **Marital Status** | | | | | | |
| Never Married | 187 | 7.9 | 223 | 10.0 | 675 | 9.2 |
| Married/ living together | 2020 | 84.9 | 1854 | 82.9 | 6147 | 83.3 |
| Divorced/separated/widowed | 173 | 7.3 | 161 | 7.2 | 558 | 7.6 |
| **Religion** | | | | | | |
| Roman catholic | 604 | 25.5 | 460 | 20.7 | 1338 | 18.2 |
| Protestant/other Christian | 1491 | 62.9 | 1495 | 67.1 | 5174 | 70.4 |
| Muslim | 209 | 8.8 | 198 | 8.7 | 627 | 8.5 |
| No religion | 66 | 2.8 | 75 | 3.4 | 209 | 2.8 |
| **Education** | | | | | | |
| No Education | 359 | 15.1 | 276 | 12.3 | 850 | 11.5 |
| Primary Incomplete | 869 | 36.5 | 727 | 32.5 | 2054 | 27.8 |
| Primary Complete | 657 | 27.6 | 686 | 30.6 | 1989 | 27.0 |
| Secondary + | 495 | 20.8 | 549 | 24.5 | 2487 | 33.7 |
| **Wealth Quintile** | | | | | | |
| Lowest | 589 | 24.8 | 543 | 24.3 | 1856 | 25.2 |
| Second | 493 | 20.7 | 443 | 19.8 | 1466 | 19.8 |
| Middle | 445 | 18.7 | 398 | 17.8 | 1333 | 18.1 |
| Fourth | 407 | 17.1 | 417 | 18.6 | 1267 | 17.2 |
| Highest | 446 | 18.7 | 437 | 19.5 | 1458 | 19.7 |
| **Province** | | | | | | |
| Nairobi | 146 | 6.1 | 127 | 5.7 | 740 | 10.0 |
| Central | 250 | 10.5 | 159 | 7.1 | 669 | 9.1 |
| Coast | 211 | 8.9 | 213 | 9.5 | 806 | 10.9 |
| Eastern | 379 | 15.9 | 331 | 14.8 | 881 | 11.9 |
| Nyanza | 355 | 14.9 | 442 | 19.7 | 1030 | 14.0 |
| Rift Valley | 667 | 28.1 | 653 | 29.2 | 2180 | 29.5 |
| Western | 308 | 13.0 | 249 | 11.2 | 841 | 11.4 |
| North Eastern | 63 | 2.6 | 64 | 2.9 | 233 | 3.2 |
| **Child's Birth Order** | | | | | | |
| 1 | 571 | 24 | 523 | 23.4 | 1964 | 26.6 |
| 2–4 | 1151 | 48.3 | 1111 | 49.7 | 3810 | 51.6 |
| 5 + | 658 | 27.7 | 603 | 26.9 | 1607 | 21.8 |

*(Continued)*

**Table 1.** (Continued)

| Variable | 2003 | | 2008 | | 2014 | |
|---|---|---|---|---|---|---|
| **Sex of child** | n (2380) | % | n (2237) | % | n (7380) | % |
| **Parity** | | | | | | |
| 0–1 | 586 | 24.6 | 533 | 23.8 | 2006 | 27.2 |
| 2–4 | 1234 | 51.9 | 1185 | 53.0 | 3925 | 53.2 |
| 5 + | 560 | 23.5 | 519 | 23.2 | 1448 | 19.6 |
| **Number of children in household** | | | | | | |
| 0–1 | 852 | 35.8 | 763 | 34.1 | 3074 | 41.6 |
| 2–4 | 1508 | 63.4 | 1430 | 63.9 | 4247 | 57.6 |
| 5 + | 21 | 0.8 | 44.1 | 2.0 | 59 | 0.8 |
| **Place of delivery** | | | | | | |
| Home | 1429 | 61.3 | 1185 | 53.9 | 2495 | 34.2 |
| Public | 572 | 24.6 | 767 | 34.9 | 3633 | 49.7 |
| Private | 329 | 14.1 | 245 | 11.2 | 1173 | 16.1 |
| **Occupation** | | | | | | |
| Unemployed | 915 | 38.5 | 954 | 42.7 | 1303 | 36.8 |
| Employed | 1463 | 61.5 | 1278 | 57.3 | 2239 | 63.2 |

in households with 2–4 children over the years. In 2014, 41.6% of households had 0–1 children, compared to 35.8% in 2003. There was an increase in deliveries taking place in public facilities over the years, accompanied by a decrease in home deliveries. In 2014, 49.7% of deliveries occurred in public facilities, compared to 61.3% in 2003. The proportion of employed mothers showed slight fluctuations over the years, indicating changes in employment rates among mothers. In 2014, 63.2% of mothers were employed, compared to 61.5% in 2003.

### Demographic and socio-economic determinants of non-vaccination amongst children 0–23 months in Kenya 2003 to 2014

In 2003, the mothers' education and Province were statistically significant. When compared to mothers with no education, the likelihood of non-vaccination was 0.55 times (AOR = 0.55, 95% CI = 0.37–0.81) among mothers who didn't complete primary education, 0.34 times (AOR = 0.34, 95% CI = 0.21–0.56) for mothers with complete primary education and 0.26 times (AOR = 0.26, 95% CI = 0.14–0.50) for mothers with secondary education or higher. Compared to children in Coast Province, the likelihood of non-vaccination were 6.04 times in Nyanza Province (AOR = 6.04, 95% CI = 2.80–13.02) and 8.60 times in North Eastern Provinces (AOR = 8.60, 95% CI = 3.36–19.18).

In 2008/09 marital status, religion, wealth quintile and Province were statistically significant for non-vaccination. Compared to mothers who have never been married, children of divorced/ separated/ widowed women were 0.22 times (AOR = 0.22, 95% CI = 0.07–0.65) likely to be non-vaccinated. Compared to children of women with no religion, those of Protestant/ Other Christian were 0.37 times (AOR = 0.37, 95% CI = 0.17–0.81) likely to be non-vaccinated. Compared to the highest wealth quintile, children of the lowest quintile were 7.3 times (AOR = 7.30, 95% CI = 2.11–25.24) and middle quintiles 4.96 (AOR = 4.96, 95% CI = 1.59–15.46) times likely to be non-vaccinated, respectively. Within the provinces, non-vaccination was 4.43 times (AOR = 4.43, 95% CI = 1.24–15.85) in Central province and 2.99 times (AOR = 2.99, 95% CI = 1.32–6.75) in North-Eastern compared to the Coast province.

In 2014, mothers age, province, birth order and place of delivery were statistically significant for non-vaccination. Non-vaccination was 12.53 times (AOR = 12.53, 95% CI = 1.59–98.73) in children whose mothers age ranged between 15–19 years compared to those aged 45–49 years. It was 7.15 times (AOR = 7.15, 95% CI = 2.02–25.30) in north-Eastern compared to Coast and 4.19 times (AOR = 4.19, 95% CI = 1.09–16.18) times in families with more than 5 children compared to those with one child. Non-vaccination was also 4.47 times (AOR = 4.47, 95% CI = 1.32–15.17) likely in children born at home compared to those born in a private facility.

Results of all analysed demographic and socio-economic determinants of non-vaccination amongst children 0–23 months in Kenya; KDHS 2003, 2008/09 and 2014 are presented in Table 2.

## Demographic and socio-economic determinants of under-vaccination amongst children 0–23 months in Kenya 2003 to 2014

In 2003, children from the Rift Valley were 0.45 times (AOR = 0.45, 95% CI = 0.25–0.82) likely to be under-vaccinated compared to children from the Coast province.

In 2008/2009, religion, Province and birth order were statistically significant. Compared to mothers with no religion children, mothers who were Roman Catholic, Protestant/ Other Christian and Muslims were 2.51 (AOR = 2.51,95% = 1.41–4.48), 2.34 (AOR = 2.34,95% CI = 1.37–3.99) and 1.99(AOR = 1.99, 95% CI = 1.07–3.69) times likely to have their children under-vaccinated, respectively. Likewise, compared to children from the Coast Province, those from Rift Valley were 0.55 times (AOR = 0.55, 95% CI = 0.36–0.86) likely to be under-vaccinated. Households with between 2–4 children were 1.42 times (AOR = 1.42, 95% CI = 1.04–1.93) likely to be under-vaccinated compared to households with 0 to 1 child.

In 2014, gender, mothers age and number of children in a household were statistically significant. Female children were 0.83 times (AOR = 0.83, 95% CI = 0.71–0.98) likely to be under-vaccinated compared to their male counterparts. Mothers aged 15–19 years were 3.27 times (AOR = 3.27, 95% CI = 1.14–9.36) likely to have under-vaccinated children compared to those aged 45–49 years. Likewise, households with between 2 to 4 children were 1.38 times (AOR = 1.38, 95% CI = 1.11–1.73) likely to be under-vaccinated compared to households with 0 to 1 child.

Results of all analysed demographic and socio-economic determinants of under-vaccination amongst children 0–23 months in Kenya; KDHS 2003, 2008/09 and 2014 are presented in Table 3.

## Demographic and socio-economic determinants of missed opportunity for vaccination amongst children 0–23 months in Kenya

In 2003, religion, Province and place of delivery were significant. Mothers who were Protestant/Other Christians were 0.46 times (AOR = 0.46, 95% CI = 0.26–0.83) likely to have children with MOV compared to those with no religion. Children living in Nyanza and Western Provinces were 2.81 (AOR = 2.81,95% I = 1.65–4.80) and 2.62 (AOR = 2.62, 95% CI = 1.62–4.26) times likely to have MOV compared to children from the Coast Province. Similarly, children delivered at home were 0.35 times (AOR = 0.35, 95% CI = 0.24–0.51) likely to have MOV compared to children born at private health facility.

In 2008/09, marital status, education, Province and birth order were statistically significant. When analysed within the marital status category, and compared to single women, the likelihood of MOV in children of those who were Married/living together and the divorced/separated/widowed was 1.64 times (AOR = 1.64, 95% CI = 1.02–2.65) and 2.01 (AOR = 2.01,95%

**Table 2. Demographic and socio-economic determinants of non-vaccination amongst children 0–23 months in Kenya; KDHS 2003, 2008/09 and 2014.**

| | 2003 (n = 2380) | | 2008/09 (n = 2237) | | 2014 (n = 7380) | |
|---|---|---|---|---|---|---|
| | COR (95% CI) | AOR[‡] (95% CI) | COR (95% CI) | AOR[‡] (95% CI) | COR (95% CI) | AOR[‡] (95% CI) |
| **Sex of child** | | | | | | |
| Male | 1 | 1 | 1 | 1 | 1 | 1 |
| Female | 0.95(0.72–1.26) | 0.98(0.73–1.330 | 0.80(0.54–1.19) | 0.82(0.56–1.21) | **1.46(1.01–2.11)** * | 1.27(0.79–2.02) |
| **Residence** | | | | | | |
| Urban | **0.43(0.27–0.68)** * | 0.87(0.46–1.66) | 0.65(0.32–1.32) | 2.57(0.98–6.72) | **0.44(0.27–0.70)** * | 1.24(0.62–2.47) |
| Rural | 1 | 1 | 1 | 1 | 1 | 1 |
| **Mother's Age** | | | | | | |
| 15–19 | **0.22(0.06–0.82)** * | 0.54(0.11–2.69) | 1.62(0.19–13.96) | 5.34(0.49–58.24) | 1.05(0.22–5.16) | **12.53(1.59–98.73)** * |
| 20–24 | **0.24(0.07–0.89)** * | 0.70(0.15–3.23) | 1.28(0.15–10.51) | 3.22(0.30–34.69) | 0.70(0.14–3.37) | 5.96(0.84–42.35) |
| 25–29 | **0.21(0.06–0.76)** * | 0.59(0.14–2.47) | 1.47(0.18–12.20) | 2.80(0.28–28.18) | 0.82(0.17–3.98) | 6.41(0.97–42.23) |
| 30–34 | **0.24(0.07–0.89)** * | 0.42(0.10–1.74) | 1.88(0.23–15.47) | 3.09(0.36–26.94) | 0.74(0.15–3.55) | 3.74(0.60–23.30) |
| 35–39 | 0.34(0.08–1.39) | 0.62(0.38–2.78) | 2.59(0.30–22.25) | 3.06(0.34–27.85) | 1.02(0.21–4.93) | 4.51(0.69–29.45) |
| 40–44 | 0.25(0.06–1.00) | 0.30(0.06–1.42) | 1.14(0.30–12.61) | 1.43(0.13–16.11) | 4.00(0.67–23.75) | 7.44(0.90–61.28) |
| 45–49 | 1 | 1 | 1 | 1 | 1 | 1 |
| **Marital Status** | | | | | | |
| Never Married | 1 | 1 | 1 | 1 | 1 | 1 |
| Married/ living together | 1.51(0.86–2.66) | 1.01(0.51–1.99) | 1.12(0.52–2.41) | 0.73(0.45–1.54) | 1.29(0.66–2.51) | 0.88(0.40–1.96) |
| Divorced/separated/widowed | 1.30(0.60–2.84) | 0.92(0.37–2.30) | 0.54(0.18–1.65) | **0.22(0.07–0.65)** * | 1.58(0.71–3.56) | 1.44(0.49–4.21) |
| **Religion** | | | | | | |
| Roman catholic | 0.48(0.22–1.07) | 0.53(0.25–1.12) | **0.39(0.16–0.94)** * | 0.46(0.20–1.05) | **0.39(0.19–0.82)** * | 1.20(0.39–3.70) |
| Protestant/other Christian | **0.45(0.20–0.98)** * | 0.53(0.26–1.08) | **0.34(0.13–0.90)** * | **0.37(0.17–0.81)** * | 0.61(0.28–1.33) | 1.21(0.36–4.02) |
| Muslim | 1.02(0.43–2.44) | 0.40(0.15–1.07) | 0.77(0.27–2.21) | 0.47(0.18–1.24) | 1.49(0.69–3.25) | 0.98(0.26–3.70) |
| No religion | 1 | 1 | 1 | 1 | 1 | 1 |
| **Education** | | | | | | |
| No Education | 1 | 1 | 1 | 1 | 1 | 1 |
| Primary Incomplete | **0.42(0.30–0.58)** * | **0.55(0.37–0.81)** * | 0.56(0.28–1.08) | 0.96(0.44–2.09) | **0.38(0.27–0.54)** * | 0.85(0.44–1.64) |
| Primary Complete | **0.20(0.13–0.31)** * | **0.34(0.21–0.56)** * | **0.45(0.22–0.92)** * | 1.32(0.57–3.07) | **0.24(0.12–0.48)** * | 0.88(0.37–2.08) |
| Secondary + | **0.14(0.08–0.23)** * | **0.26(0.14–0.50)** * | **0.31(0.14–0.69)** * | 1.14(0.47–2.76) | **0.18(0.11–0.30)** * | 0.55(0.18–1.69) |
| **Wealth Quintile** | | | | | | |
| Lowest | **4.70(2.78–7.95)** * | 1.31(0.56–3.04) | **4.08(1.72–9.68)** * | **7.30(2.11–25.24)** * | **7.19(2.45–21.16)** * | 1.42(0.31–6.53) |
| Second | **1.98(1.13–3.46)** * | 0.80(0.34–1.86) | 1.46(0.57–3.75) | 2.75(0.83–9.08) | 2.66(0.87–8.20)* | 0.67(0.144–3.11) |
| Middle | 1.74(0.96–3.15) | 0.95(0.40–2.27) | **2.61(1.04–6.59)** * | **4.96(1.59–15.46)** * | **3.74(1.08–12.98)** * | 0.79(0.18–3.43) |
| Fourth | 1.05(0.57–1.92) | 0.68(0.30–1.58) | 1.13(0.41–3.12) | 2.10(0.73–6.06) | 1.39(0.41–4.70) | 0.99(0.23–4.28) |
| Highest | 1 | 1 | 1 | 1 | 1 | 1 |
| **Province** | | | | | | |
| Nairobi | 0.48(0.21–1.06) | 1.42(0.51–4.00) | 1.20(0.33–4.37) | 4.26(0.88–20.70) | 0.42(0.06–3.23) | NA |
| Central | 0.67(0.32–1.38) | 1.81(0.72–4.55) | 1.20(0.45–3.17) | **4.43(1.24–15.85)** * | **0.19(0.06–0.63)** * | 0.92(0.14–6.01) |
| Coast | 1 | 1 | 1 | 1 | 1 | 1 |
| Eastern | 0.68(0.34–1.36) | 1.10(0.47–2.55) | 0.78(0.32–1.86) | 1.03(0.39–2.75) | 0.47(0.17–1.31) | 1.08(0.28–4.19) |
| Nyanza | **3.28(1.72–6.27)** * | **6.04(2.80–13.02)** * | 1.42(0.70–2.89) | 2.37(0.95–5.90) | 1.17(0.54–2.52) | 1.63(0.41–6.56) |
| Rift Valley | 1.34(0.71–2.52) | 1.67(0.84–3.36) | 0.91(0.39–2.13) | 0.99(0.40–2.41) | **2.15(1.10–4.18)** * | 2.62(0.80–8.55) |
| Western | 1.65(0.90–3.03) | 2.49(1.22–5.05) | 1.48(0.57–3.88) | 2.18(0.68–6.91) | 1.89(0.73–4.88) | 1.68(0.43–6.54) |
| North Eastern | **12.95(6.69–25.07)** * | **8.60(3.86–19.18)** * | **5.13(2.30–11.45)** * | **2.99(1.32–6.75)** * | **7.76(3.80–15.84)** * | **7.15(2.02–25.30)** * |
| **Child's Birth Order** | | | | | | |
| 1 | 1 | 1 | 1 | 1 | 1 | 1 |
| 2–4 | 1.08(0.74–1.57) | 1.66(0.67–4.11) | 1.16(0.64–2.09) | 1.12(0.50–2.49) | 1.03(0.64–1.65) | 1.36(0.56–3.27) |

*(Continued)*

**Table 2.** (Continued)

| | 2003 (n = 2380) | | 2008/09 (n = 2237) | | 2014 (n = 7380) | |
|---|---|---|---|---|---|---|
| | COR (95% CI) | AOR‡ (95% CI) | COR (95% CI) | AOR‡ (95% CI) | COR (95% CI) | AOR‡ (95% CI) |
| 5 + | **2.19(1.55–3.10)** * | 1.60(0.47–5.42) | **2.86(1.48–5.54)** * | 1.61(0.49–5.28) | **2.89(1.66–5.01)** * | **4.19(1.09–16.18)** * |
| **Parity** | | | | | | |
| 0–1 | 1 | 1 | 1 | 1 | 1 | 1 |
| 2–4 | 0.89(0.63–1.26) | 0.46(0.18–1.17) | 1.28(0.72–2.29) | 1.10(0.48–2.51) | 1.19(0.73–1.94) | 0.48(0.21–1.08) |
| 5 + | **2.07(1.47–2.90)** * | 0.95(0.30–3.04) | **2.95(1.49–5.87)** * | 1.97(0.56–6.98) | **3.29(1.85–5.83)** * | 0.36(0.99–1.34) |
| **Number of children in household** | | | | | | |
| 0–1 | 1 | 1 | 1 | 1 | 1 | 1 |
| 2–4 | 1.31(0.98–1.76) | 1.09(0.75–1.56) | **2.77(1.72–4.44)** * | **1.98(1.07–3.69)** * | **2.21(1.47–3.33)** * | 1.43(0.74–2.74) |
| 5 + | 0.96(0.29–3.26) | 1.16(0.37–3.64) | 1.67(0.33–8.45) | 1.40(0.28–7.01) | 0.95(0.18–5.01) | NA |
| **Place of delivery** | | | | | | |
| Home | **3.24(1.74–6.03)** * | 1.85(0.99–3.47) | 1.75(0.63–4.84) | 1.21(0.29–5.10) | **6.26(1.98–19.87)** * | **4.47(1.32–15.17)** * |
| Public | 0.84(0.40–1.76) | 0.92(0.44–1.92) | 0.55(0.19–1.58) | 0.50(0.13–2.03) | 1.35(0.40–4.52) | 1.17(0.33–4.14) |
| Private | 1 | 1 | 1 | 1 | 1 | 1 |
| **Occupation** | | | | | | |
| Unemployed | 1 | 1 | 1 | 1 | 1 | 1 |
| Employed | 0.82(0.60–1.12) | 0.78(0.57–1.05) | 0.73(0.48–1.13) | 0.79 (0.49–1.26) | **0.57(0.38–0.85)** * | 0.88(0.57–1.36) |

*P-value <0.05

‡ **Adjusted Odds Ratio (AOR):** All variables from the bivariable logistic regression were included in the multivariable logistic regression model using the enter method selection criteria

N/A- fewer observations hence omitted in the regression model

CI = 1.03–3.92), respectively. Similarly, within education category, when compared to mothers with no education, the likelihood of MOV among those with incomplete primary education was 0.57 (AOR = 0.57, 95% CI = 0.38–0.84), 0.57 (AOR = 0.57, 95% CI = 0.36–0.90) amongst those with complete primary education and 0.47 (AOR = 0.47, 95% CI = 0.28–0.79) amongst mothers with secondary education and above. When compared to children from Coast Province, the likelihood of MOV amongst children from Eastern Province was 0.39 (AOR = 0.39, 95% CI = 0.22–0.69) and 0.45 (AOR = 0.45, 95% CI = 0.25–0.80) in Rift-Valley Province. Children in the 5+ birth order were 2.82 times (AOR = 2.82, 95% CI = 1.17–6.84) likely to experience MOV compared to first-borns.

In 2014, wealth quintile and Province were statistically significant. Similarly, as compared to Coast Province, children from Western and North-Eastern Provinces were 1.67 times (AOR = 1.67, 95% CI = 1.04–2.67) and 0.54 times (AOR = 0.54, 95% CI = 0.31–0.93) likely to have MOV.

Results of all analysed demographic and socio-economic determinants of MOV amongst children 0–23 months in Kenya; KDHS 2003, 2008/09 and 2014 are presented in Table 4.

## Discussion

Our findings have shown the influence of various demographic and socio-economic determinants of non-vaccination, under-vaccination and MOV amongst children 0–23 months in Kenya from 2003 to 2014.

With regards to non-vaccination one of the relevant factors included the mothers' education. A similar findings was reported in another study, with maternal education standing out

**Table 3. Demographic and socio-economic determinants of under vaccination among children 0–23 months in Kenya; KDHS 2003, 2008/09 and 2014.**

| Variables | 2003 (n = 2380) | | 2008/09 (n = 2237) | | 2014 (n = 7380) | |
|---|---|---|---|---|---|---|
| | COR (95% CI) | AOR‡ (95% CI) | COR (95% CI) | AOR‡ (95% CI) | COR (95% CI) | AOR‡ (95% CI) |
| **Sex of child** | | | | | | |
| Male | 1 | 1 | 1 | 1 | 1 | 1 |
| Female | 1.03(0.85–1.25) | 1.02(0.83–1.25) | 0.88(0.68–1.13) | 0.89(0.69–1.16) | 0.96(0.85–1.09) | **0.83(0.71–0.98)** * |
| **Residence** | | | | | | |
| Urban | 1.22(0.94–1.58) | 1.52(0.93–2.47) | 0.71(0.48–1.05) | 0.72(0.46–1.13) | 0.89(0.78–1.01) | 1.07(0.86–1.33) |
| Rural | 1 | 1 | 1 | 1 | 1 | |
| **Mother's Age** | | | | | | |
| 15–19 | 2.03(0.54–7.67) | 2.08(0.48–9.02) | 1.35(0.47–3.88) | 1.78(0.57–5.62) | **2.08(1.02–4.24)** * | **3.27(1.14–9.36)** * |
| 20–24 | 1.44(0.38–5.41) | 1.26(0.29–5.47) | 1.32(0.47–3.70) | 1.48(0.50–4.43) | 1.44(0.72–2.90) | 2.05(0.78–5.42) |
| 25–29 | 1.47(0.39–5.54) | 1.15(0.27–4.93) | 1.21(0.42–3.44) | 1.18(0.41–3.42) | 1.25(0.63–2.51) | 1.66(0.64–4.36) |
| 30–34 | 1.46(0.38–5.57) | 1.09(0.25–4.67) | 0.79(0.27–2.33) | 0.67(0.23–2.00) | 1.35(0.67–2.70) | 1.85(0.71–4.77) |
| 35–39 | 1.25(0.32–4.83) | 0.97(0.23–4.13) | 0.94(0.32–2.74) | 0.75(0.25–2.28) | 1.66(0.81–3.37) | 1.93(0.73–5.05) |
| 40–44 | 1.20(0.29–4.88) | 0.95(0.22–4.17) | 1.20(0.37–3.91) | 0.99(0.31–3.15) | 1.21(0.55–2.68) | 2.04(0.74–5.64) |
| 45–49 | 1 | 1 | 1 | 1 | 1 | 1 |
| **Marital Status** | | | | | | |
| Never Married | 1 | 1 | 1 | 1 | 1 | 1 |
| Married/living together | 0.95(0.69–1.30) | 0.95(0.65–1.38) | 0.86(0.61–1.23) | 0.85(0.55–1.32) | 0.90(0.73–1.10) | 1.04(0.75–1.44) |
| Divorced/separated/widowed | 1.13(0.72–1.77) | 1.10(0.68–1.79) | 1.18(0.73–1.91) | 1.15(0.65–2.02) | 0.74(0.54–1.00) | 0.79(0.49–1.27) |
| **Religion** | | | | | | |
| Roman catholic | 0.99(0.59–1.66) | 0.97(0.56–1.69) | 1.52(0.98–2.35) | **2.51(1.41–4.48)** * | 1.12(0.80–1.55) | 1.22(0.77–1.92) |
| Protestant/other Christian | 0.93(0.56–1.56) | 0.96(0.55–1.69) | 1.29(0.82–2.01) | **2.34(1.37–3.99)** * | 1.14(0.84–1.56) | 1.23(0.79–1.92) |
| Muslim | 0.85(0.48–1.52) | 1.17(0.60–2.29) | 1.49(0.90–2.48) | **1.99(1.07–3.69)** * | 1.09(0.77–1.54) | 0.10(0.66–1.84) |
| No religion | 1 | 1 | 1 | 1 | 1 | 1 |
| **Education** | | | | | | |
| No Education | 1 | 1 | 1 | 1 | 1 | 1 |
| Primary Incomplete | 1.20(0.95–1.53) | 0.96(071–1.31) | 0.89(0.57–1.38) | 0.66(0.37–1.16) | 0.89(0.74–1.07) | 0.89(0.66–1.21) |
| Primary Complete | 1.10(0.86–1.40) | 0.94(0.69–1.29) | 0.79(0.53–1.20) | 0.63(0.37–1.05) | **0.75(0.62–0.91)** * | 0.85(0.62–1.18) |
| Secondary+ | 0.99(0.75–1.31) | 0.90(0.62–1.32) | 0.75(0.49–1.17) | 0.72(0.43–1.22) | **0.68(0.57–0.82)** * | 0.92(0.66–1.29) |
| **Wealth Quintile** | | | | | | |
| Lowest | 0.90(0.68–1.19) | 1.07(0.65–1.75) | 1.46(0.94–2.28) | 1.27(0.78–2.07) | **1.49(1.20–1.84)** * | 1.39(0.92–2.09) |
| Second | 1.20(0.90–1.60) | 1.36(0.84–2.21) | **1.63(1.05–2.53)** * | 1.37(0.84–2.24) | 1.22(0.98–1.51) | 0.98(0.67–1.44) |
| Middle | 0.83(0.62–1.10) | 0.98(0.61–1.58) | 1.27(0.79–2.04) | 1.13(0.68–1.86) | 1.13(0.89–1.43) | 1.12(0.77–1.64) |
| Fourth | 1.06(0.78–1.43) | 1.28(0.80–2.06) | 1.28(0.80–2.04) | 1.09(0.69–1.73) | 1.02(0.79–1.33) | 1.07(0.73–1.57) |
| Highest | 1 | 1 | 1 | 1 | 1 | 1 |
| **Province** | | | | | | |
| Nairobi | 1.16(0.82–1.62) | 1.23(0.76–1.97) | 0.99(0.66–1.48) | 1.35(0.78–2.35) | 0.88(0.64–1.22) | 0.94(0.56–1.59) |
| Central | 0.79(0.57–1.09) | 1.09(0.73–1.64) | 0.77(0.48–1.23) | 0,73(0.43–1.25) | 0.69(0.53–0.90) | 0.90(0.57–1.44) |
| Coast | 1 | 1 | 1 | 1 | 1 | 1 |
| Eastern | 1.08(0.80–1.44) | 1.36(0.91–2.04) | 0.94(0.67–1.31) | 0.74(0.47–1.17) | 0.92(0.73–1.17) | 1.12(0.75–1.66) |
| Nyanza | 1.12(0.80–1.57) | 1.34(0.87–2.06) | 1.26(0.92–1.71) | 0.96(0.63–1.47) | 0.98(0.80–1.20) | 1.15(0.78–1.71) |
| Rift Valley | 1.10(0.80–1.50) | 1.31(0.89–1.93) | **0.67(0.47–0.98)** * | **0.55(0.36–0.86)** * | 1.07(0.89–1.28) | 1.16(0.82–1.63) |
| Western | 1.14(0.82–1.59) | 1.33(0.90–1.97) | 0.97(0.63–1.50) | 0.80(0.47–1.39) | 0.89(0.72–1.11) | 0.96(0.65–1.43) |
| North Eastern | 0.50(0.30–0.83) | **0.45(0.25–0.82)** * | 0.89(0.55–1.44) | 0.59(0.29–1.22) | 0.92(0.69–1.23) | 0.95(0.60–1.51) |
| **Child's Birth Order** | | | | | | |
| 1 | 1 | 1 | 1 | 1 | 1 | |
| 2–4 | 1.16(0.93–1.45) | 1.28(0.68–2.02) | 1.21(0.94–1.55) | 1.65(0.74–3.69) | 1.11(0.96–1.29) | 0.77(0.38–1.55) |

*(Continued)*

**Table 3.** (Continued)

| Variables | 2003 (n = 2380) | | 2008/09 (n = 2237) | | 2014 (n = 7380) | |
|---|---|---|---|---|---|---|
| | COR (95% CI) | AOR‡ (95% CI) | COR (95% CI) | AOR‡ (95% CI) | COR (95% CI) | AOR‡ (95% CI) |
| 5+ | 1.04(0.81–1.34) | 1.80(0.81–4.04) | 1.21(0.89–1.64) | 2.52(0.87–7.39) | **1.33(1.11–1.60)** * | 0.98(0.40–2.36) |
| **Parity** | | | | | | |
| 0–1 | 1 | 1 | 1 | 1 | 1 | 1 |
| 2–4 | 1.18(0.96–1.46) | 1.06(0.56–2.02) | 1.20(0.93–1.54) | 0.81(0.35–1.89) | 1.06(0.91–1.23) | 1.32(0.64–2.72) |
| 5+ | 0.97(0.74–1.27) | 0.72(0.30–1.72) | 1.12(0.82–1.54) | 0.68(0.22–2.06) | **1.30(1.08–1.56)** * | 1.37(0.55–3.38) |
| **Number of children in household** | | | | | | |
| 0–1 | 1 | 1 | 1 | 1 | 1 | 1 |
| 2–4 | 1.19(0.98–1.43) | 1.17(0.93–1.48) | **1.61(1.22–2.14)** * | **1.42(1.04–1.93)** * | **1.37(1.21–1.56)** * | **1.38(1.11–1.73)** * |
| 5+ | 1.23(0.61–2.48) | 1.42(0.66–3.08) | 1.94(0.88–4.25) | 1.37(0.65–2.84) | 1.44(0.87–2.40) | 0.74(0.43–1.27) |
| **Place of delivery** | | | | | | |
| Home | 1.02(0.76–1.37) | 1.02(0.73–1.41) | 0.97(0.65–1.44) | 0.77(0.49–1.21) | **1.35(1.10–1.67)** * | 1.20(0.87–1.65) |
| Public | 0.86(0.61–1.20) | 0.85(0.59–1.20) | 0.87(0.56–1.34) | 0.72(0.44–1.16) | **1.32(1.08–1.61)** * | 1.37(0.71–1.49) |
| Private | 1 | 1 | 1 | 1 | 1 | 1 |
| **Occupation** | | | | | | |
| Unemployed | 1 | 1 | 1 | 1 | 1 | 1 |
| Employed | 0.93(0.77–1.11) | 0.95(0.77–1.16) | 0.95(0.78–1.16) | 0.97(0.77–1.23) | 0.91(0.76–1.08) | 1.00(0.81–1.22) |

*P-value <0.05

‡ **Adjusted Odds Ratio (AOR):** All variables from the bivariable logistic regression were included in the multivariable logistic regression model using the enter method selection criteria

N/A- fewer observations hence omitted in the regression model

as a crucial factor influencing childhood vaccination [22]. Mothers with lower education may have reduced awareness, lower health literacy, and lack access to healthcare resources, which can negatively influence their children's vaccination status. there is evidence showing that mothers lacking post-secondary education are more prone to overlooking their children's vaccination schedules [23]. Furthermore, a positive correlation exists between maternal education and childhood immunization [24]. This connection becomes particularly significant as mothers with primary education possess essential health knowledge, while those with secondary education and beyond exhibit the necessary communication skills to advocate for child immunization [24]. In Mozambique, for example, children of uneducated mothers demonstrate lower vaccine uptake [25]. Consequently, mothers with lower literacy levels may be less inclined to actively participate in community health programs and initiatives, comprehend the risks and benefits associated with vaccinations, exert limited influence within their social networks, and face challenges in making empowered decisions, particularly regarding the vaccination status of their children. The correlation between lower literacy and reduced engagement with healthcare information may contribute to a lack of awareness about the importance of vaccinations, potentially hindering these mothers from fully embracing vaccination for their children.

In this study, across the 3 surveys, regional disparities have been noted for non-vaccination. Regional clustering and variations in non-vaccinations have also been observed in other studies [26]. It is therefore important to identify factors that influence these regional differences. The differences could be due to individual, contextual and systematic factors including socioeconomic inequalities in vaccine uptake in these regions [27]. These have also been described for Kenyan communities, especially those living in the refugee communities [28]. In Nyanza

**Table 4. Demographic and socio-economic determinants of missed opportunity for vaccination amongst children 0–23 months in Kenya; KDHS 2003, 2008/09 and 2014.**

| Variable | 2003 (n = 2380) | | 2008/09 (n = 2237) | | 2014 (n = 7380) | |
|---|---|---|---|---|---|---|
| | COR (95% CI) | AOR‡ (95% CI) | COR (95% CI) | AOR‡ (95% CI) | COR (95% CI) | AOR‡ (95% CI) |
| **Sex of child** | | | | | | |
| Male | 1 | 1 | 1 | 1 | 1 | 1 |
| Female | 0.87(0.70–1.08) | 0.85(0.68–1.07) | 0.88(0.69–1.11) | 0.90(0.71–1.15) | **0.88(0.78–0.99)** * | 1.00(0.84–1.18) |
| **Residence** | | | | | | |
| Urban | 1.13(0.87–1.47) | 1.17(0.75–1.83) | **0.63(0.42–0.95)** * | 0.62(0.36–1.07) | **0.71(0.61–0.82)** * | 0.88(0.70–1.10) |
| Rural | 1 | 1 | 1 | 1 | 1 | 1 |
| **Mother's Age** | | | | | | |
| 15–19 | 1.42(0.17–11.81) | 0.78(0.08–7.00) | 1.00(0.31–3.22) | 3.32(0.97–11.39) | 1.50(0.73–3.08) | 1.11(0.38–3.29) |
| 20–24 | 1.25(0.15–10.06) | 0.71(0.08–6.03) | 1.17(0.37–3.68) | 3.33(1.03–10.78) | 1.34(0.68–2.66) | 1.20(0.45–3.19) |
| 25–29 | 1.14(0.14–9.10) | 0.75(0.09–6.19) | 0.92(0.30–2.88) | 2.32(0.73–7.34) | 1.23(0.61–2.46) | 1.08(0.41–2.82) |
| 30–34 | 1.15(0.14–9.26) | 0.68(0.08–5.52) | 1.03(0.31–3.39) | 2.19(0.69–6.91) | 1.15(0.58–2.30) | 1.06(0.42–2.68) |
| 35–39 | 0.98(0.12–8.08) | 0.69(0.09–5.47) | 0.93(0.26–3.35) | 1.55(0.47–5.14) | 1.12(0.55–2.27) | 1.10(0.43–2.84) |
| 40–44 | 0.71(0.08–6.71) | 0.56(0.06–5.29) | 1.27(0.35–4.60) | 2.06(0.61–6.99) | 1.38(0.66–2.90) | 1.35(0.49–3.75) |
| 45–49 | 1 | 1 | 1 | 1 | 1 | 1 |
| **Marital Status** | | | | | | |
| Never Married | 1 | 1 | 1 | 1 | 1 | 1 |
| Married/living together | 1.09(0.72–1.64) | 1.20(0.78–1.84) | **1.63(1.11–2.39)** * | **1.64(1.02–2.65)** * | 1.05(0.86–1.29) | 1.33(0.95–1.86) |
| Divorced/separated/widowed | 0.87(0.50–1.53) | 1.08(0.58–2.01) | **2.00(1.13–3.52)** * | **2.01(1.03–3.92)** * | 0.93(0.70–1.24) | 1.08(0.69–1.69) |
| **Religion** | | | | | | |
| Roman catholic | 0.87(0.47–1.62) | 0.58(0.31–1.08) | 0.84(0.47–1.50) | 2.23(1.20–4.16) | 0.78(0.53–1.15) | 0.98(0.57–1.69) |
| Protestant/other Christian | 0.74(0.41–1.33) * | **0.46(0.26–0.83)** * | 0.84(0.49–1.43) | 2.05(1.12–3.73) | 0.87(0.62–1.23) | 1.11(0.68–1.80) |
| Muslim | 0.63(0.33–1.21) | 0.70(0.37–1.33) | 1.01(0.59–1.75) | 1.55(0.84–2.85) | 0.55(0.37–0.81) | 1.18(0.62–2.25) |
| No religion | 1 | 1 | 1 | 1 | 1 | 1 |
| **Education** | | | | | | |
| No Education | 1 | 1 | 1 | 1 | 1 | 1 |
| Primary Incomplete | **1.51(1.08–2.13)** * | 1.06(0.68–1.65) | **0.71(0.53–0.95)** * | **0.57(0.38–0.84)** * | **1.43(1.15–1.77)** * | 1.07(0.75–1.520 |
| Primary Complete | 1.22(0.83–1.79) | 0.87(0.53–1.42) | **0.56(0.40–0.78)** * | **0.57(0.36–0.90)** * | 1.16(0.93–1.44) | 1.04(0.70–1.53) |
| Secondary+ | 1.31(0.90–1.91) | 0.73(0.44–1.22) | **0.39(0.28–0.56)** * | **0.47(0.28–0.79)** * | 1.17(0.95–1.45) | 1.01(0.68–1.51) |
| **Wealth Quintile** | | | | | | |
| Lowest | 0.83(0.60–1.16) | 0.99(0.56–1.72) | **2.05(1.33–3.16)** * | 1.07(0.54–2.15) | **1.43(1.15–1.78)** * | 1.65(1.12–2.43) |
| Second | 1.09(0.77–1.53) | 1.23(0.71–2.15) | **1.78(1.07–2.94)** * | 0.99(0.50–1.97) | **1.60(1.28–1.99)** * | 1.30(0.90–1.88) |
| Middle | 0.77(0.53–1.11) | 0.92(0.54–1.58) | 1.56(0.97–2.51) | 0.96(0.49–1.88) | **1.40(1.11–1.77)** * | 1.13(0.79–1.61) |
| Fourth | 0.77(0.54–1.11) | 0.94(0.57–1.55) | 1.29(0.80–2.10) | 0.96(0.52–1.76) | **1.31(1.02–1.68)** * | 1.30(0.92–1.85) |
| Highest | 1 | 1 | 1 | 1 | 1 | 1 |
| **Province** | | | | | | |
| Nairobi | 1.34(0.87–2.05) | 1.05(0.62–1.80) | 0.75(0.45–1.25) | 1.40(0.73–2.68) | 0.64(0.43–95) | 1.04(0.59–1.82) |
| Central | 0.88(0.57–1.34) | 0.82(0.48–1.41) | **0.41(0.23–0.73)** * | 0.53(0.26–1.08) | 0.83(0.61–1.12) | 1.14(0.70–1.84) |
| Coast | 1 | 1 | 1 | 1 | 1 | 1 |
| Eastern | 0.95(0.61–1.46) | 1.09(0.65–1.83) | 0.46(0.30–0.72) * | **0.39(0.22–0.69)** * | 1.06(0.81–1.37) | 1.47(0.94–2.30) |
| Nyanza | **2.36(1.53–3.62)** * | **2.81(1.65–4.80)** * | 1.11(0.75–1.65) | 1.03(0.60–1.78) | 1.21(0.96–1.52) | 1.30(0.85–2.00) |
| Rift Valley | 1.14(0.81–1.62) | 1.25(0.80–1.96) | 0.59(0.35–1.00) | **0.45(0.25–0.80)** * | 1.02(0.82–1.28) | 1.48(0.99–2.23) |
| Western | **2.13(1.46–3.10)** * | **2.62(1.62–4.26)** * | 1.61(1.04–2.50) * | 1.46(0.80–2.66) | **1.44(1.12–1.87)** * | **1.67(1.04–2.67)** * |
| North Eastern | 0.55(0.28–1.10) | 0.63(0.30–1.30) | 0.90(0.52–1.57) | 0.50(0.25–1.00) | **0.39(0.30–0.53)** * | **0.54(0.31–0.93)** * |
| **Child's Birth Order** | | | | | | |
| 1 | 1 | 1 | 1 | 1 | 1 | 1 |

*(Continued)*

**Table 4.** (Continued)

| Variable | 2003 (n = 2380) | | 2008/09 (n = 2237) | | 2014 (n = 7380) | |
|---|---|---|---|---|---|---|
| | COR (95% CI) | AOR‡ (95% CI) | COR (95% CI) | AOR‡ (95% CI) | COR (95% CI) | AOR‡ (95% CI) |
| 2–4 | 0.93(0.71–1.22) | 1.42(0.75–2.70) | 1.08(0.80–1.45) | 1.18(0.63–2.22) | **0.83(0.73–0.96)** * | 1.02(0.52–2.00) |
| 5+ | 0.77(0.54–1.09) | 1.06(0.44–2.53) | **1.78(1.24–2.57)** * | **2.82(1.17–6.84)** * | 1.00(0.84–1.18) | 1.38(0.59–3.20) |
| **Parity** | | | | | | |
| 0–1 | 1 | 1 | 1 | 1 | 1 | 1 |
| 2–4 | 0.84(0.64–1.10) | 0.72(0.37–1.39) | 1.14(0.85–1.53) | 0.76(0.41–1.41) | **0.82(0.72–0.95)** * | 0.80(0.40–1.58) |
| 5+ | 0.76(0.54–1.08) | 0.98(0.40–2.37) | **1.65(1.11–2.46)** * | 0.59(0.24–1.46) | 0.97(0.82–1.15) | 0.58(0.25–1.36) |
| **Number of children in household** | | | | | | |
| 0–1 | 1 | 1 | 1 | 1 | 1 | 1 |
| 2–4 | 0.82(0.65–1.02) | 0.90(0.68–1.20) | **1.49(1.12–1.98)** * | 1.05(0.78–1.41) | 1.12(0.99–1.26) | 1.10(0.88–1.38) |
| 5+ | 0.68(0.24–1.96) | 0.70.0(0.22–2.30) | **4.77(1.78–12.79)** * | 2.14(0.78–1.41) | 0.81(0.38–1.73) | 0.46(0.16–1.132) |
| **Place of delivery** | | | | | | |
| Private | 1 | 1 | 1 | 1 | 1 | 1 |
| Public | 0.71(0.51–1.00) | 0.75(0.53–1.07) | 0.81(0.52–1.25) | 0.74(0.45–1.21) | 1.12(0.92–1.36) | 1.03(0.75–1.41) |
| Home | **0.45(0.32–0.62)** * | **0.35(0.24–0.51)** * | **1.74(1.14–2.65)** * | 1.37(0.84–2.25) | 0.88(0.72–1.08) | 0.78(0.54–1.12) |
| **Occupation** | | | | | | |
| Unemployed | 1 | 1 | 1 | 1 | 1 | 1 |
| Employed | **1.36(1.08–1.72)** * | 1.22(0.94–1.59) | 0.96(0.78–1.18) | 0.98(0.77–1.26) | **1.24(1.04–1.48)** * | 1.19(0.98–1.45) |

*P-value <0.05

‡ **Adjusted Odds Ratio (AOR):** All variables from the bivariable logistic regression were included in the multivariable logistic regression model using the enter method selection criteria

N/A- fewer observations hence omitted in the regression model

and Western Provinces, for example, there are cultural and religious practices and beliefs that have been identified to influence non-vaccination. These include religious sects that don't believe in vaccination as well as cultural practices that prohibit vaccination and access to general health care [29].

Maternal age and family dynamics play pivotal roles in determining the likelihood of non-vaccination among children. Specifically, children born to mothers aged 15–19 years and those within larger families encountered significantly elevated odds of remaining unvaccinated. Additionally, children born to younger mothers and those in the 5+ birth order were prone to non-vaccination compared to their counterparts, particularly when contrasted with first-born children. This trend has been reported in a different setting [9] and could be related to the complacency and lack of focus that comes with maternal experience as well as having many children. Some reasons may be related to women being fully engaged with domestic work, and hence they tend to forget their children's vaccination timing [30].

The socio-economic status of mothers or caretakers directly impacts on their ability to finance health requirements [31]. In our study, when we compared to the highest wealth quintile, children of the lowest quintile and middle quintiles were likely to be non-vaccinated. There is evidence indicating that children belonging to poor households were most likely to have fewer interactions with immunization services leading to non-vaccination [32]. Families in economically-disadvantaged households encounter challenges in accessing immunization services due to financial constraints, limited awareness, insufficient healthcare infrastructure, educational barriers, demanding work schedules, potential fear or mistrust of healthcare systems, and geographic inaccessibility among other reasons. These factors collectively contribute

to a heightened prevalence of non-vaccination, as families grapple with issues such as transportation costs, lack of information, scarce resources, educational limitations, work-related hurdles, and apprehensions regarding the effectiveness and safety of vaccines.

The birthing environment influences non-vaccination in children through factors such as access to healthcare services, health education, economic status, cultural beliefs, healthcare infrastructure, and government policies. In settings with limited healthcare access or inadequate health education, parents may lack awareness of the importance of vaccinations, contributing to non-compliance. Economic challenges and cultural influences prevalent in the birthing environment can further impact vaccination decisions. The availability and quality of healthcare infrastructure, as well as government policies supporting immunization programs, also play important roles. As observed in this study, children born at home were more likely to be non-vaccinated as compared to those born in private facilities. This observation is aligned with evidence from a different setting that reported significant advantages conferred to children born in health facilities, including vaccination [30].

Regarding under-vaccination, there were varied demographic and socio-economic determinants reported amongst children 0–23 months in Kenya from 2003 to 2014.

The current study demonstrated that the number of children in a household influenced a child's under-vaccination status. Our study findings have shown that in 2008/09 period, households with between 2–4 children were likely to be under-vaccinated compared to households with 0 to 1 child. Similarly, in 2014, households with between 2 to 4 children were likely to be under-vaccinated compared to households with 0 to 1 child. A similar finding was reported from a different setting where the number of siblings in a family directly impacted on the mother's ability to spare time to bring another sibling to the health facility for vaccination [31]. The presence of other siblings was determined to be an independent predictor for under-vaccination [33]. Similar in Bangladesh, children born in families with three or more siblings had a reduced probability of being vaccinated [34].

There is a global natal inequality where boys receive preferential treatment in access and utilization of services than girls as they grow up. In our study, in 2014, the female children were likely to be under-vaccinated compared to their male counterparts. Similar findings have been reported in a different part of the world where girls are less likely to be vaccinated than boys [35]. Similar gender inequities in immunization coverage were found to be prevalent even for individual level vaccine antigens [36]. The higher likelihood of male children being taken to the hospital, as opposed to their female counterparts, can be attributed to cultural perceptions emphasizing family lineage continuation by boys, in contrast to the expectation that girls will marry into other families, particularly prevalent in many underdeveloped societies where the birth of a male child is esteemed and carries greater societal value [37].

Province had a significant influence on under-vaccination in Kenya. In 2003 and 2008/09, children from the Rift-Valley were likely to be under-vaccinated compared to children from the coast. The Rift-Valley is such an expansive region with sparsely populated areas and infrastructural difficulties related to access to immunization services. It has been reported that less than 50% of the total population live less than one-hour to a health facility [38]. This is most likely to influence immunization decisions taken by mothers and ultimately impact in under-vaccination status.

In our study, mothers aged 15–19 years were likely to have under-vaccinated children compared to those aged 45–49 years. Other studies have reported that older women (35 years and above) were more likely to take their children for basic vaccination [39].

Our findings also demonstrate that religion significantly affected under-vaccination. Compared to mothers with no religion, children whose mothers were Roman Catholic, Protestant/Other Christian and Muslims were likely to have their children under-vaccinated. A study in

the region similarly reported that children belonging to Roman catholic mothers were less likely to vaccinate their children as compared to those belonging to orthodox churches [40]. It is likely that a multifaceted interplay of factors, encompassing specific religious beliefs, the sway of religious leaders, educational and socioeconomic gaps within religious communities, geographic and cultural influences, and prevailing community norms would influence the utilisation of health amenities in the community, including child immunization services. Disparities in vaccination rates among religious groups often arise from differing levels of acceptance rooted in religious teachings, potential scepticism voiced by religious leaders, and the intricate social dynamics within these communities. In Kenya, for instance, the Catholic Church has raised concerns about certain vaccine antigens, potentially shaping the overall vaccination practices of its members [41]. These instances underscore the need for nuanced and culturally sensitive approaches to vaccination promotion within religious communities, recognizing and addressing the unique influences that shape health-related decisions.

There were varied demographic and socio-economic determinants of MOV that were reported amongst children 0–23 months in Kenya from 2003 to 2014.

First, our findings conflict others that reported no influence of the mothers marital status on the child's MOV status [25, 42]. In our study, there were higher odds of MOV in children of married women and those in relationships as compared to mothers who have never been married. This may stem from shared decision-making dynamics within relationships. In marriages and partnerships, discussions around childcare responsibilities, work schedules, and conflicting priorities might lead to missed opportunities for timely vaccinations. Single mothers, on the other hand, often bear the primary responsibility for their child's healthcare decisions, potentially fostering a greater sense of individual agency and commitment to preventive measures. Additionally, differing levels of health awareness, access to healthcare services, and the distribution of caregiving responsibilities within marital or relationship contexts could contribute to variations in the likelihood of seizing vaccination opportunities.

There is evidence to show that there is a relationship between wealth and missed opportunities [7]. In 2014, the study revealed that women in the lowest wealth quintile were more likely to have their children missing opportunity for vaccination as compared to those in the highest wealth quintile. This may be explained by differences in priorities, where women in the lowest wealth quintile may have different priorities from those in the highest quintile given their socio-economic needs. Vaccination for their children, given they are not manifesting any signs or symptoms of illness, may rank lower to them compared to fending for their daily needs. Similarly, for the same reasons, households with more than five children were more likely to have MOV compared to those with 1 child.

In our study, children delivered at home were likely to have MOV compared to those born in private health facility. Similar findings have been reported in other studies [9, 43] where children born in health facilities were significantly more likely to be vaccinated and up to date with their vaccine schedules as compared to children born at home. Given the limited access to healthcare infrastructure associated with home deliveries, it is likely to potentially result in MOV. The preference for home deliveries might be associated with lower health literacy or socio-economic challenges among mothers, impacting their awareness and ability to prioritize timely immunizations [44–46]. In contrast, children born in private health facilities are likely to receive immediate postnatal care, including vaccinations, in a controlled and supervised environment.

Children living in Nyanza and Western Provinces were likely to have MOV compared to children from the Coast Province in 2003. These regional differences could also be due to individual, contextual and systematic factors including socioeconomic inequalities in vaccine uptake [27]. Similar to those noted for non-vaccination above, these factors have also been

described for Kenyan communities, especially those of the Somali community and living in the refugee communities [28]. Cultural and religious practises and beliefs in Western Province may influence non-vaccination. These include religious sects that don't believe in vaccination as well as cultural practises that prohibit vaccination and access to general health care [29].

## Limitations

This study utilized data from the KDHS conducted between 2003–2014, to determine the influence of demographic and socio-economic factors on non-vaccination, under-vaccination and MOV amongst children 0–23 months in Kenya. Relying on existing data limits the inclusion of detailed variables and may introduce errors from the original data collection. The binary categorization of vaccination status, determined by an algorithm, might oversimplify the intricate nature of immunization patterns, and the cross-sectional design prevents establishing cause-and-effect relationships. Additionally, potential biases, like social desirability in self-reported data, and unobserved social and cultural factors, may affect the reliability of the results. Changes in healthcare policies and contextual factors over time, along with missing data and ethical considerations, further contribute to the study's limitations. While we have endeavoured to mitigate these limitations through rigorous analytical methods and transparent reporting, we recognize that these limitations and biases remain inherent to our study design. It is therefore important to bear these in mind while interpreting our findings, and also within the context of these constraints. These offer avenues for future research or enhancements in data collection methods that may address these limitations.

## Conclusion

Non-vaccination was influenced by factors such as education, province, and place of delivery in 2003, while marital status, religion, wealth quintile, and birth order played a pivotal role in 2008/09. Maternal education emerged as a crucial factor affecting childhood vaccination, with regional disparities indicating the impact of cultural and religious practices. Maternal age and family dynamics were identified as significant contributors to non-vaccination risk. The socio-economic status of mothers, reflected in wealth quintile, directly affected vaccination outcomes.

Under-vaccination determinants varied over the years, involving factors such as province, religion, gender, mother's age, and household size. Regional differences, particularly in Nyanza and Western Provinces, highlighted the influence of cultural and religious beliefs on under-vaccination. The global gender inequality observed with access to health services was observed, with female children less likely to be under-vaccinated.

Missed opportunities for vaccination (MOV) exhibited varied determinants, including marital status, education, province, birth order, wealth quintile, and place of delivery. Unique findings included single mothers having fully vaccinated children, contrary to previous studies. Wealth disparities and larger households were associated with increased MOV risk, while home births were linked to a higher likelihood of missed opportunities. Regional differences persisted, especially in Nyanza and Western Provinces.

## Recommendations

In order to improve childhood vaccination in Kenya, it will be important to have tailored educational campaigns by implementing targeted maternal education programs to raise awareness about the importance of childhood vaccination. The focus should be on dispelling myths and addressing concerns to improve overall vaccine acceptance.

Region-specific interventions are critical in the development and implementation of strategies to address cultural and religious influences on vaccination decisions. Interventions should be tailored to the unique challenges observed in provinces like Nyanza and Western Provinces.

There is value in evaluating the implementation of economic support for vulnerable families by providing targeted support and subsidies for families in lower wealth quintiles to ensure equitable access to vaccines. This could involve making vaccines more affordable and accessible for economically disadvantaged families.

Gender-equal vaccination initiatives should be implemented to ensure equal access to vaccination for both male and female children. Awareness campaigns should specifically target communities where gender disparities in vaccination coverage exist.

Integrated Missed Opportunities Interventions should be developed and implemented. Comprehensive interventions should target factors associated with missed opportunities for vaccination, including marital status, education, wealth quintile, and place of delivery. Healthcare facilities should be strengthened to capitalize on every interaction for vaccination promotion and administration. To accurately assess the magnitude of Missed Opportunities for Vaccination (MOV), a process to verify home-based vaccination records used in surveys should be considered during data collection. This can help identify any discrepancies and provide more reliable data for future analyses.

Finally, further qualitative research is needed to gain a deeper understanding of the contextual factors influencing vaccination decisions, access to vaccines, and the operational aspects of service delivery points. These studies can provide valuable insights into the specific challenges faced by communities and inform the development of tailored interventions.

## Supporting information

**S1 Checklist. STROBE statement—Checklist of items that should be included in reports of observational studies.**
(DOCX)

## Acknowledgments

We appreciate and acknowledge the children and their caregivers for providing the KDHS data and the DHS program for the public availability of data that enabled this analysis.

## Author Contributions

**Conceptualization:** Christopher Ochieng' Odero.

**Data curation:** Christopher Ochieng' Odero.

**Formal analysis:** Christopher Ochieng' Odero, Vincent Omondi Were.

**Methodology:** Christopher Ochieng' Odero, Doreen Othero, Vincent Omondi Were, Collins Ouma.

**Validation:** Christopher Ochieng' Odero, Doreen Othero, Collins Ouma.

**Writing – original draft:** Christopher Ochieng' Odero.

**Writing – review & editing:** Christopher Ochieng' Odero, Doreen Othero, Vincent Omondi Were, Collins Ouma.

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
