## [Decision Letter · Decision Letter 0]

22 Feb 2024

PGPH-D-24-00200

The influence of demographic and socio-economic factors on non-vaccination, under-vaccination and Missed Opportunities for Vaccination amongst children 0-23 months in Kenya for the period 2003-2014.

Dear Dr. Odero,

Thank you for submitting your manuscript to PLOS Global Public Health. After careful consideration, we feel that it has merit but does not fully meet PLOS Global Public Health’s publication criteria as it currently stands. Therefore, we invite you to submit a revised version of the manuscript that addresses the points raised during the review process.

We look forward to receiving your revised manuscript.

Kind regards,

Collins Otieno Asweto, PhD

Academic Editor

Journal Requirements:

Additional Editor Comments (if provided):

Reviewers' comments:

Reviewer's Responses to Questions

**Comments to the Author**

1. Does this manuscript meet PLOS Global Public Health’s publication criteria? Is the manuscript technically sound, and do the data support the conclusions? The manuscript must describe methodologically and ethically rigorous research with conclusions that are appropriately drawn based on the data presented.

Reviewer #1: Yes

Reviewer #2: Partly

Reviewer #3: Yes

Reviewer #4: Yes

2. Has the statistical analysis been performed appropriately and rigorously?

Reviewer #1: Yes

Reviewer #2: No

Reviewer #3: Yes

Reviewer #4: Yes

3. Have the authors made all data underlying the findings in their manuscript fully available (please refer to the Data Availability Statement at the start of the manuscript PDF file)?

Reviewer #1: Yes

Reviewer #2: Yes

Reviewer #3: Yes

Reviewer #4: Yes

4. Is the manuscript presented in an intelligible fashion and written in standard English?

Reviewer #1: Yes

Reviewer #2: Yes

Reviewer #3: Yes

Reviewer #4: Yes

5. Review Comments to the Author

Reviewer #1: I found the manuscript to be well-written, comprehensive, and methodologically sound. The authors have conducted a thorough investigation into demographic and socio-economic determinants of immunization coverage gaps in Kenya. The research question is clearly articulated and addresses an important issue in public health. The methodology is rigorous and appropriate for the research objectives, and the results are well-presented and supported by the data analysis. Also, the discussion section provides insightful interpretations of the findings and contextualizes them within the existing literature. However, there is a need for citation at the end of line 416: “The preference for home deliveries might be associated with lower health literacy or socio-economic challenges among mothers, impacting their awareness and ability to prioritize timely immunizations” because evidence exists. Overall, the manuscript adheres to the journal's guidelines and standards for publication.

Reviewer #2: This is a promising article in an important topic with potential for application to public health policies in Kenya. However, in its current format, the manuscript does not meet the criteria for a scientific publication. The inexperience of the researchers is apparent in several aspects, from the design of the study to the quality of the manuscript. This article would benefit from a major review by a experienced epidemiologist; commenting in all the areas where the manuscript benefits from improvement is beyond the scope of this review. Some of the fundamental flaws, which are not addressed, are obviously apparent. For example, why did the authors not combine the three waves of DHS, which is the standard and preferable approach when analysis DHS data? Combining the three waves of DHS would increased the sample size, therefore increasing power to detect smaller effects, and the inclusion of an indicator variable for the wave would indicate temporal trends in the outcome, from one wave of DHS to the other. The "methods" section is superficial and does not describe, nor address, fundamental questions related to a epidemiological study such as this one. The absence of relevant references is present throughout the manuscript. Overall, this manuscript would benefit from the collaboration with an experienced epidemiologist who could point out the extensive opportunities for improvement.

Reviewer #3: This was a quantitative study that analyzed secondary data from longitudinal repeated annual and national cross-sectional surveys obtained from the Kenya demographic and health surveys conducted in 2003-2014. These data were analyzed to determine Non-vaccination, under-vaccination and missed opportunities for vaccination which had posed significant challenges to the benefits of vaccination in children 0-23months in Kenya.

The authors have presented an insightful study from a massive research with results, statistics and analysis performed to a high standard with sufficient details presented. The conclusions drawn were supported by the data presented in the research work. I commend the authors for a diligent and painstaking research work presented here. It captures the essence of a good quantitative study.

Reviewer #4: The immunization rates in Kenya throughout the time period under study were expertly highlighted by the writers. However, I have made the following observations that need correction:

1. There were numerous typographical errors that need attention. See, for instance, pages 222, 280, 365, 405, etc. In several lines, there were also missing punctuation marks. A thorough editing will address this oversight.

2. The writers used only one citation when they wrote, "...other studies cite that..." The wording should indicate if they are relying just on one citation to support their claim.

3. The language syntax used in the results presentation should be adjusted to accurately represent the message that the authors intended to convey. Example, lines 187–193 may cause confusion for the reader on the relationship between the mother's primary school education and non-vaccination.

6. PLOS authors have the option to publish the peer review history of their article (what does this mean?). If published, this will include your full peer review and any attached files.

**Do you want your identity to be public for this peer review?** For information about this choice, including consent withdrawal, please see our Privacy Policy.

Reviewer #1: **Yes: **Grace Ibor

Reviewer #2: No

Reviewer #3: **Yes: **PRISCILIA UHUANMWEN IMADE

Reviewer #4: No

---

## [Decision Letter · Decision Letter 1]

27 Mar 2024

PGPH-D-24-00200R1

The influence of demographic and socio-economic factors on non-vaccination, under-vaccination and Missed Opportunities for Vaccination amongst children 0-23 months in Kenya for the period 2003-2014.

Dear Odero,

Thank you for submitting your manuscript to PLOS Global Public Health. After careful consideration, we feel that it has merit but does not fully meet PLOS Global Public Health’s publication criteria as it currently stands. Therefore, we invite you to submit a revised version of the manuscript that addresses the points raised during the review process.

We look forward to receiving your revised manuscript.

Kind regards,

Collins Otieno Asweto, PhD

Academic Editor

Journal Requirements:

2. We have noticed that you have uploaded Supporting Information files, but you have not included a list of legends. Please add a full list of legends for your Supporting Information files after the references list.

Additional Editor Comments (if provided):

Reviewers' comments:

Reviewer's Responses to Questions

**Comments to the Author**

1. If the authors have adequately addressed your comments raised in a previous round of review and you feel that this manuscript is now acceptable for publication, you may indicate that here to bypass the “Comments to the Author” section, enter your conflict of interest statement in the “Confidential to Editor” section, and submit your "Accept" recommendation.

Reviewer #5: All comments have been addressed

Reviewer #6: (No Response)

Reviewer #7: (No Response)

2. Does this manuscript meet PLOS Global Public Health’s publication criteria? Is the manuscript technically sound, and do the data support the conclusions? The manuscript must describe methodologically and ethically rigorous research with conclusions that are appropriately drawn based on the data presented.

Reviewer #5: Yes

Reviewer #6: Yes

Reviewer #7: Yes

3. Has the statistical analysis been performed appropriately and rigorously?

Reviewer #5: Yes

Reviewer #6: Yes

Reviewer #7: Yes

4. Have the authors made all data underlying the findings in their manuscript fully available (please refer to the Data Availability Statement at the start of the manuscript PDF file)?

Reviewer #5: Yes

Reviewer #6: Yes

Reviewer #7: Yes

5. Is the manuscript presented in an intelligible fashion and written in standard English?

Reviewer #5: Yes

Reviewer #6: Yes

Reviewer #7: Yes

6. Review Comments to the Author

Reviewer #5: Materials and methodology

Strength:

- The sample size calculation is comprehensive, considering various factors like design effect and response rates, ensuring robust estimation.

Recommendations:

- Address potential biases and limitations in the methodology to strengthen validity.

- Provide contextual interpretation of results and ensure consistency in reporting across different years for better comparison.

Results:

- Tables are well-arranged and interpreted, demonstrating meticulous attention to detail and enhancing the study's credibility.

Discussion:

- Clear, accurate, and insightful discussion on factors influencing childhood vaccination in Kenya.

Reviewer #6: Introduction:

1. It would be good to add similar studies conducted within the region(Talk about non-vaccination, under vaccination and Missed Opportunities for Vaccination in East African countries, if there are any.

2. Add the reference for this paragraph>>>>In Kenya, the National Vaccines and Immunization Program (NVIP) recommends that a child receives Bacillus Calmette–Guérin (BCG) and oral polio vaccine (OPV) at birth; pentavalent vaccine, OPV, rotavirus vaccine and pneumococcal conjugate vaccine (PCV) at 6, 10 and 14 weeks, vitamin A at 6 months and measles vaccine at 9 months. Routine vaccines are delivered for free at public health facilities, with outreach services conducted in hard-to-reach communities. Through routine vaccination the government aims to achieve equitable access to health services, especially for children.

Methods

Why were all the variables from the bivariate logistic regression included in the multivariate logistic regression? As it is better to include only those that are statistically significant, it would be good if you could provide/add a clear explanation of why you have chosen to include all the variables.

Results

In Table 3, the variable Protestant/other Christian has been made significant for the 2003 data while the CI crosses 1: 0.74 (0.41-1.33). Is it a mistake or there is a reason behind?

Reviewer #7: Very interesting research topic: classic but complex due to the number of outcome variables (non-vaccination, under-vaccination and missed opportunities vaccination) and the number of survey repetitions (2003, 2008/09 and 2014).

The study has been conducted rigorously and the parallelism of forms has been respected between the periods.

The authors have methodically integrated the observations of my peer reviewers, which significantly improved the quality of the first manuscript. However, the current version of the manuscript would gain more if the authors could consider the additional comments below:

Methods section (2/2):

1. We suggest the authors to explain how the Littoral province was chosen as the reference modality for the Province variable (It’s not necessary to explain the choice of reference modality for each of the other variables).

2. The study area needs to be completed with more details. Indeed, the manuscript specifies: “Study area This study covered the entire 47 counties in Kenya. The KDHS surveys were national level surveys, which in 2003 and 2008/09 were conducted across all the former eight Provinces of Kenya and in 2014, was conducted in all the 47 counties in line with the new administrative units”. This means changes in the administrative division between 2009 and 2014. If so, the authors are invited to explain or specify that the changes in the administrative division have not affected the provinces (in number and name) between 2003 and 2014.

Results section (2/2):

1. We recommend that the authors present the socio-demographic characteristics of the study participants in the form of tables and/or graphs (numbers, frequencies, means, standard deviation, median, etc. depending on the characteristic and/or the outcome variable).

2. We know that for a given variable, OR (reference modality) = 1. I suggest that the authors replace REF by 1 in each table (Tables 1, 2 and 3).

Another comment (1/1):

1. I didn't see any appendices in the current version of the manuscript. If you have not already done so, it would be advisable to include the following: the bivariable logistic regression and the multivariable logistic regression model initial.

7. PLOS authors have the option to publish the peer review history of their article (what does this mean?). If published, this will include your full peer review and any attached files.

**Do you want your identity to be public for this peer review?** For information about this choice, including consent withdrawal, please see our Privacy Policy.

Reviewer #5: **Yes: **Sarashwati Giri

Reviewer #6: No

Reviewer #7: **Yes: **Lazare M'BOUNGOU

---

## [Decision Letter · Decision Letter 2]

10 May 2024

The influence of demographic and socio-economic factors on non-vaccination, under-vaccination and Missed Opportunities for Vaccination amongst children 0-23 months in Kenya for the period 2003-2014.

PGPH-D-24-00200R2

Dear Odero,

We are pleased to inform you that your manuscript 'The influence of demographic and socio-economic factors on non-vaccination, under-vaccination and Missed Opportunities for Vaccination amongst children 0-23 months in Kenya for the period 2003-2014.' has been provisionally accepted for publication in PLOS Global Public Health.

Best regards,

Collins Otieno Asweto, PhD

Academic Editor

Reviewer's Responses to Questions

**Comments to the Author**

1. If the authors have adequately addressed your comments raised in a previous round of review and you feel that this manuscript is now acceptable for publication, you may indicate that here to bypass the “Comments to the Author” section, enter your conflict of interest statement in the “Confidential to Editor” section, and submit your "Accept" recommendation.

Reviewer #5: All comments have been addressed

Reviewer #7: All comments have been addressed

2. Does this manuscript meet PLOS Global Public Health’s publication criteria? Is the manuscript technically sound, and do the data support the conclusions? The manuscript must describe methodologically and ethically rigorous research with conclusions that are appropriately drawn based on the data presented.

Reviewer #5: Yes

Reviewer #7: Yes

3. Has the statistical analysis been performed appropriately and rigorously?

Reviewer #5: Yes

Reviewer #7: Yes

4. Have the authors made all data underlying the findings in their manuscript fully available (please refer to the Data Availability Statement at the start of the manuscript PDF file)?

Reviewer #5: Yes

Reviewer #7: Yes

5. Is the manuscript presented in an intelligible fashion and written in standard English?

Reviewer #5: Yes

Reviewer #7: Yes

6. Review Comments to the Author

Reviewer #5: all comments are addressed

Reviewer #7: None.

7. PLOS authors have the option to publish the peer review history of their article (what does this mean?). If published, this will include your full peer review and any attached files.

**Do you want your identity to be public for this peer review?** For information about this choice, including consent withdrawal, please see our Privacy Policy.

Reviewer #5: **Yes: **Sarashwati Giri

Reviewer #7: **Yes: **Lazare M'BOUNGOU
